# RAGE against the Machine: Can Increasing Our Understanding of RAGE Help Us to Battle SARS-CoV-2 Infection in Pregnancy?

**DOI:** 10.3390/ijms23126359

**Published:** 2022-06-07

**Authors:** Courtney K. Kurashima, Po’okela K. Ng, Claire E. Kendal-Wright

**Affiliations:** 1School of Natural Sciences and Mathematics, Chaminade University of Honolulu, Honolulu, HI 96816, USA; kurashima@student.chaminade.edu (C.K.K.); spencerkamakane@gmail.com (P.K.N.); 2Department of Obstetrics, Gynecology and Women’s Health, John A. Burns School of Medicine, University of Hawai’i, Honolulu, HI 96813, USA; 3Department of Anatomy, Biochemistry and Physiology, John A. Burns School of Medicine, University of Hawai’i, Honolulu, HI 96813, USA

**Keywords:** RAGE, pregnancy, SARS-CoV-2, inflammation, minorities, Hawai’i

## Abstract

The receptor of advanced glycation end products (RAGE) is a receptor that is thought to be a key driver of inflammation in pregnancy, SARS-CoV-2, and also in the comorbidities that are known to aggravate these afflictions. In addition to this, vulnerable populations are particularly susceptible to the negative health outcomes when these afflictions are experienced in concert. RAGE binds a number of ligands produced by tissue damage and cellular stress, and its activation triggers the proinflammatory transcription factor Nuclear Factor Kappa B (NF-κB), with the subsequent generation of key proinflammatory cytokines. While this is important for fetal membrane weakening, RAGE is also activated at the end of pregnancy in the uterus, placenta, and cervix. The comorbidities of hypertension, cardiovascular disease, diabetes, and obesity are known to lead to poor pregnancy outcomes, and particularly in populations such as Native Hawaiians and Pacific Islanders. They have also been linked to RAGE activation when individuals are infected with SARS-CoV-2. Therefore, we propose that increasing our understanding of this receptor system will help us to understand how these various afflictions converge, how forms of RAGE could be used as a biomarker, and if its manipulation could be used to develop future therapeutic targets to help those at risk.

## 1. Introduction

A healthy term pregnancy is dependent on the interplay of numerous factors, including those that influence an individual’s genetic background; how maternal tissues interact with those of the fetus; how cellular pathways progress throughout gestation to eventually drive parturition; and how the environment shapes exogenous chemical exposure through nutrition, habitat, culture, and the experience of pregnancy [1,2]. Unfortunately, despite the burden that this presents to new mothers, they now have to also navigate the ongoing severe acute respiratory syndrome coronavirus 2 (SARS-CoV-2) pandemic during their pregnancies. This virus has indiscriminately functioned like a ‘machine’ to spread over the entire globe and impact everyone. However, over the last two years, it has become clear SARS-CoV-2 infection [3]. Very rapidly, it was understood that several factors, such as childhood obesity, diabetes (DM), asthma, chronic lung disease, sickle cell disease, cardiovascular (CVD) disease, or being immunocompromised or over 65, increase the risk of severe illness from the SARS-CoV-2 virus [4]. In addition to this, it has been established that, if you are pregnant, or were recently pregnant, you are also more likely to suffer from severe SARS-CoV-2 symptoms [5]. You are also at an increased risk of complications that can affect your pregnancy and the developing fetus [5]. Thus, as many of these SARS-CoV-2 comorbidities, either with or without pregnancy, are exacerbated by socioeconomic status [6], this new virus has also amplified and exposed the gross inequities in healthcare for minorities and those of lower socioeconomic status (Figure 1), disproportionately impacting these communities [7].

Biologically and pathophysiologically, the conditions of pregnancy and viral infection intersect at many levels, including presenting challenges to the maternal immune system [8]. Inflammation and its regulation are central to implantation [9]; however, its relative suppression in the tissues of pregnancy is necessary throughout the majority of gestation to help the mother tolerate the genetically different fetus [10]. By contrast, at the end of pregnancy, the initiation of inflammation is crucial for successful parturition and the timely delivery of the fetus [11]. Approximately 40% of preterm births (PTB) are caused by intrauterine infection and its resultant inflammatory cascade [12,13]. In these cases, the infectious agent promotes a profuse inflammatory response by seizing the immune system and its cells, which ultimately terminates with the prompt eviction of the baby [13]. Thus, inflammation is central to both the normal mechanisms of parturition and deliveries caused by infection.

One receptor that has been shown to be key for the detection of various ligands, and that is known to be activated at the end of pregnancy, is the receptor of advanced glycation end products (RAGE) [14]. It interacts with danger-associated molecular patterns (DAMPs), which are generated by cellular stress [15], and advanced glycation end products (AGE), which are produced by the combination of fat or protein with sugars in the bloodstream [16,17]. The activation of RAGE results in the activation of the proinflammatory transcription factors Nuclear Factor Kappa B (NF-κB) and Activator Protein 1 (AP1), and oxidative-stress-response genes [18,19,20]. Therefore, as inflammation cascades are central to SARS-CoV-2 pathophysiology, many studies have focused on the potential role of RAGE in this viral infection [21]. Indeed, it is thought that, during infection, when inflammation is active, RAGE could also be activated by a slew of pertinent comorbidities, which contribute to the inflammatory load and the cytokine storm that are so frequently described with severe SARS-CoV-2 infection [22]. Thus, as RAGE is so frequently activated in both pregnancy and the crucial comorbidities that have been described to intensify SARS-CoV-2, this may explain why they increase the risk for severe forms of disease and provide both a biomarker to direct more accurate prognosis or treatment strategies, and future putative intervention targets.

## 2. The Receptor for Advanced Glycation End Products

The receptor for advanced glycation end products (RAGE) is a pattern-recognition receptor that belongs to the immunoglobulin superfamily of cell surface receptors. These proteins contain at least one immunoglobulin domain or fold, and are generally associated with cell adhesion, binding, and recognition. More specifically, the members of the immunoglobulin superfamily of cell surface receptors play pivotal roles in molecular transport, toxin neutralization, and pattern recognition, and as cell phenotype markers, cell adhesion molecules, and viral receptors [23]. The involvement of RAGE in the progression of inflammatory and neurodegenerative disorders has been well studied. Inflammation plays a central role in many physiological processes that are important in the progression of diabetes, lung diseases, and fetal membrane weakening, which are characterized by elevated levels of RAGE in circulation [19,24,25]. RAGE exerts its physiological and pathological effects through its ability to bind with a multitude of ligands. These include several of the DAMPs, such as high-mobility group box 1 (HMGB1), certain S100 proteins, Amyloid β, DNA and RNA, and AGE [15]. However, the specific signaling responses that are precipitated by ligand binding not only depend on the specific ligand/RAGE pairing [26], but also on the cell type that it is expressed on. Generally, RAGE activation leads to downstream inflammation, and once bound to its ligand, RAGE oligomerization can trigger multiple signaling cascades and the activation of NF-κB. In addition to the normal role of RAGE associated with innate immunity, it can also orchestrate cellular proliferation and differentiation [27]. Thus, increases in the RAGE ligands have also been implicated in a range of disease states, such as Alzheimer’s Disease, diabetes, and during tumor growth [19,28,29].

### 2.1. RAGE Structure and Forms

The structure of RAGE consists of three different extracellular domains (V, C, C’), a transmembrane region, and a cytosolic domain from which it transduces signals into the cell. This full-length (fl-RAGE) form remains embedded in the membrane through the anchoring of the transmembrane region, giving the receptor the ability to transduce communications both intra- and extracellularly. In contrast, its soluble forms (cRAGE, esRAGE) consist of the three extracellular domains only, and these collectively are often referred to as sRAGE. The formation of these receptor forms can occur through alternative splicing (esRAGE), as well as by protease cleavage by the membrane metalloproteinase ADAM10 (cRAGE). When liberated enzymatically, the separation of the extracellular portion from the intracellular domain is modulated by Ca^2+^, allowing it to circulate freely in tissues and the vascular system. The remaining membrane-bound domain is then degraded via γ-secretase [27]. The balance of the RAGE species is dictated by the amount of proteolytic cleavage; when it increases, the sRAGE levels are high and fl-RAGE decreases, but when cleavage is inhibited, fl-RAGE is high and sRAGE is low [27]. When the soluble forms are released, not only are they capable of lowering the pool of available fl-RAGE, but they also act similarly to an inhibitor for RAGE by binding and sequestering its ligands. This neutralizes it’s signaling as the available ligands for fl-RAGE are decreased [30].

### 2.2. Expression of RAGE Varies across Tissues with Age

With the exception of the early developmental stages, RAGE expression is low or nonexistent in the majority of all tissues. Thus, in adults, the skin and lungs appear to be the only tissues in which RAGE is expressed at significant levels under normal conditions, where it is found on the resident alveolar epithelial cells and keratinocytes [22,27,31]. However, during age-related diseases and/or chronic inflammatory conditions, such as diabetes mellitus (DM) and acute respiratory distress syndrome (ARDS), circulating levels of sRAGE, as well as RAGE ligands, are altered in a manner that correlates with the severity of the associated condition. It has also been shown that RAGE ligands (i.e., HMGB1, AGEs) accumulate in tissues as part of the normal aging process, as well as in response to oxidative stress [32,33,34,35]. Interestingly, AGE accumulation is also considered a source of oxidative stress, further promoting AGE accumulation in a feedforward manner [27,32].

### 2.3. RAGE Signaling in the Fetal Membranes, Placenta, and Uterus

Increasingly, it is becoming accepted that RAGE signaling plays a significant role in driving the inflammation that leads to the rupture of the human fetal membranes (FMs) [20,25]. Indeed, this outcome is dependent on the sophisticated communication between the uterus, placenta, and FM, as RAGE signal transduction flows from maternal to fetal tissues, and back again. Although distinctly separate tissues, their collaborative influence on pregnancy outcomes are key to gestational health. Indeed, they all display patterns of inflammatory cytokine involvement in the initiation of labor and cervical ripening. Thus, inflammation is necessary for many of the key parturition pathways, including fetal membrane weakening, regardless of whether it occurs at the end of normal gestation or preterm (<37 weeks of gestation). RAGE signaling is influential for both preterm and full-term pregnancies’ FM weakening and rupture via its ligand HMGB1 [20,25]. Its binding to RAGE has been identified as an activator of NF-κB, which leads to a release of inflammatory cytokines, such as Interleukin-6 (IL6), Granulocyte Macrophage Stimulating Factor (GM-CSF), and Tumor Necrosis Factor Alpha (TNF-α). All of these are key cytokines that directly weaken the FM [11] by increasing the activation of the key matrix metalloproteinases (MMP) MMP-9 and MMP-3, which degrade the extracellular matrix (ECM). The ECM-rich amnion maintains the strength of the membranes, and thus the degeneration of its structural integrity eventually causes it to rupture [36,37]. This is important, as FM rupture can often not only help to initiate labor at term, but the premature rupture of this tissue can also lead to PTB and/or the increased risk of infection in the intrauterine compartment. RAGE activation has also been documented in the placenta, uterus, and cervix, illustrating how key both this receptor and inflammation cascades are when pregnancy nears its end and the tissues prepare for the delivery of the fetus.

RAGE activation by its ligands has been implicated in many negative pregnancy outcomes (Table 1). These include preeclampsia, hypertension, gestation diabetes, and PTB. Infection during pregnancy is one of the most important scenarios in which RAGE is activated. This is because infection is the leading cause of PTB, accounting for nearly 50% of all cases [38]. Infection can either be transmitted by vaginal flora ascending and colonizing the FM and then the intraamniotic compartment, or through placental transmission from the mother’s vasculature to the cord, and then the fetus. Regardless of the route, this leads to inflammation that often results in the rapid delivery of the fetus. In these instances, the inflammation that is caused by the infectious agent generates RAGE ligands, and this contributes to the proinflammatory milieu.

Infections such as chorioamnionitis and IAI put pregnant women at significant risk from negative pregnancy outcomes, even when they do not have SARS-CoV-2. This is because they can lead to inflammation-induced PTB and PE or other delivery complications [39,40,41,42,43]. It is known that, during these infections, RAGE and its ligands (for example, HMGB1 and S100 ß protein) play a role in the inflammatory response, although this is, as yet, not fully understood. RAGE’s counterpart, sRAGE, is also known to play a role, but it is anti-inflammatory during pregnancy infection [41,43,44,45]. The proinflammatory and anti-inflammatory RAGE ligands are found across many different tissues of pregnancies (including the AF, FM, and placenta (Table 1)) in an infection, and these further serve to increase proinflammatory cytokines, such as IL-6, IL-1ß, and TNFα, [40,45]. Thus, these RAGE inflammatory pathways are thought to be important during chorioamnionitis or IAI, but the exact mechanisms and potential for their future therapies are still yet to be deciphered [45]. Adding the further complication of a SARS-CoV-2 diagnosis to a pregnancy with another infection present could potentially combine to exacerbate these outcomes (Figure 1). Therefore, further research into these infections in tandem with SARS-CoV-2 will provide important insight into how to reduce and prevent their associated negative outcomes, which have become apparent during the pandemic for pregnant mothers.
ijms-23-06359-t001_Table 1Table 1Articles focused on RAGE, AGE, and other RAGE ligands in pregnancy.ClinicalContextGeneralized OutcomeSpeciesTissue or Cell TargetPreeclampsiaIncreased expression/level of 2AGEs, RAGE, and other RAGE ligandsHumanPlacenta [43,46,47,48,49,50],Maternal Peripheral Blood [39,47,51], Maternal Serum [14,52,53,54,55], AF [14], Umbilical Blood [47], Myometrium [56], Extravillous Trophoblasts [57], Cord Blood [14],Syncytiotrophoblast [58],Primary Cultured Adipocytes [59]
Heparin’s anti-inflammatory effect on HMGB1/RAGE axis in PEHumanPlacenta [35]
Overview of AGE, RAGE, and its signaling molecules in multiple tissues (review)
[60,61]Preeclampsia TreatmentEpigallocatechin gallate as a potential treatment to downregulate AGE–RAGE signaling pathway
Genomics [62]Hypertensive DisorderIncreased expression/level of AGEs, RAGE, and other RAGE and inflammatory ligandsHumanPlacenta [63],Primary Cultured Adipocytes [59]GestationalDiabetesIncreased level of AGEs, RAGE, and other RAGE ligandsHumanPlacenta [46], Umbilical Cord [64], Plasma [64]

In vitroUmbilical Vein Endothelial Cells [65]
Association with circular RNAsHumanPlacenta [66]
AGEs, RAGE, and RAGE ligands as both anti- and proinflammatory mediatorsHumanFM [67], Omental Adipose TissueExplants [67], Serum [67]
RAGE gene polymorphisms (review)
Gene Expression [68]
RAGE clinical opinions for treatment and management (review)
[69]GestationalDiabetesTreatmentUrsolic acid and fetal developmental defectsRatPlacenta [70]
sRAGE as a potential protective moleculeRatFetus [71]GestationalDiabetes ScreeningPotential biomarkers
[72]
AGE and RAGE levels remained unchanged, suggesting oral glucose-tolerance tests are safe for pregnant womenHumanMaternal Serum [73]DiabetesRAGE knockout mice and diabetic embryopathyMouseMaternal Plasma [74]
AGEs, sRAGE, and proinflammatory cytokine pregnancyHumanPlasma [75]
RAGE and AGE signaling in diabetic pregnancy (review)HumanMyometrium [76]DiabetesTreatmentToxicity of N-Epsilon-(carboxymethyl)lysine and bioaffinity to RAGEIn vitroUmbilical Vein Endothelial Cells [77]Preterm BirthIncreased expression/levels of AGE, RAGE, and RAGE ligandsHumanAF [40,78,79], Cervix [80], FM [81], Placenta [20]
Decreased sRAGEHumanMaternal Serum [82,83],Plasma [84], Maternal Blood [85]
Germ-free fetal pigs could be a favorable model to study immunocompromised preterm infantsPigFetus [86]
Description of RAGE, TLRs, and NF-kB in inflammatory pathways (review)
[87]Preterm LaborChanges in inflammatory signaling molecules, DAMPs, and RAGE (review)
[88]
Identified multiple AF proteins (including enRAGE) that were associated with women in threatened preterm laborHumanAF [89]PretermPrematureRupture of theMembranessRAGE, HMGB1, and AGE levelsHumanPlasma and Serum ExtractedExtracellular vesicles [90]
Increased HMGB1 and decreased sRAGE levels in clinical chorioamnionitisHumanAF [91,92]
RAGE increases with cigarette smoke condensateHumanFM [93]
FM weakening in pPROM and the mechanisms of inflammation in RAGE and NLRP7 inflammasome (review)HumanFM [25]PrematureRupture of the MembranesIncreased levels of sRAGE and esRAGEHumanPlasma [94]CervicalInsufficiencyIdentified potential biomarkers for PTB in cervical insufficiency, including enRAGE, S100A8/A9HumanAF [95,96]InfectionduringPregnancyChorioamnionitis–sRAGE expression decreased in airways and circulationHumanHuman Fetal Tracheobronchial Aspirate Fluid [41]
Increased expression/level of AGE, RAGE, and RAGE ligands in IAIHumanAF [97], Placenta [40], FM [45]

PigAF [98]
RAGE inhibition protects against fetal weight loss during secondhand-smoke-induced IUGRMouseMouse Trophoblast Cells [44]
AGEs and HMGB1 could promote sterile inflammation via monocytes/macrophagesIn vitroPlacental Cells [99]
RAGE/NF-KB pathway can increase the risk of placental vascular permeabilityIn vitroBeWo Cells [42]
Increased HMGB1 expression/levels correlates with URSAHumanFM [100,101]
Increased expression in S100 proteins in RAGE receptor binding of patients with HBVHumanPlacenta [102]
Identified genomic instabilities in pregnancy complication, which were potentially due to defective DNA on trophoblast cells and a possible RAGE-mediated mechanismHumanPlacenta [103]GeneralRAGE signaling throughout gestationHumanFM [104]
AGE/RAGE and focal adhesion that may contribute to COPD
Computer Model [105]
Increased levels of sRAGE are associated with recurrent pregnancy lossHumanBlood [106]
Secondhand smoke exposure increases RAGEMouseFetal Lung [107]
RAGE upregulation via retinol
[108]
RAGE and parturition (review)
[109]Abbreviations: Amniotic fluid (AF); advanced glycation end product (AGE); chronic obstructive pulmonary disease (COPD); danger-associated molecular patterns (DAMPs); endogenous soluble receptor for advanced glycation end products (esRAGE); fetal membrane (FM); gestational diabetes (GD); hepatitis B virus (HBV); high-mobility group box 1 (HMGB1); hypertensive disorder (HD); intra-amniotic infection (IAI); intrauterine growth restriction (IUGR); Nuclear Factor Kappa B (NF-KB); preeclampsia (PE); premature rupture of the membranes (PROM); preterm birth (PTB); preterm premature rupture of the membranes (pPROM); receptor for advanced glycation end products (RAGE); rupture of the membranes (ROM); soluble form of RAGE (sRAGE); toll-like receptor (TLR); unexplained recurrent spontaneous abortion (URSA).


## 3. SARS-CoV-2

The SARS-CoV-2 virus primarily leads to respiratory symptoms, although infected individuals can also suffer from headaches, gastrointestinal (GI) issues, fevers, fatigue, and muscle aches [110]. This virus leads to death in 0.1–25% of infected people, depending on the country [111]. Even if individuals recover from these acute symptoms, 10% of people infected with SARS-CoV-2 have also suffered from what has been termed ‘long COVID’, where they experience fatigue or dyspnea from four weeks to six months [112,113]. However, this virus is continually evolving, and it has mutated into multiple variants, with the variants of most concern to date being the B.1.617.2 (Delta) and B.1.1.529 (Omicron) [114,115]. These variants have caused an increase in transmissibility, increased risk of severe outcomes (i.e., hospitalizations), and a reduced treatment efficacy compared to the originally discovered virus form [115]. This has also made these SARS-CoV-2 variants responsible for an increased risk of mortality in unvaccinated individuals [4].

Coronaviruses are a part of the family of coronaviridae, which is an enveloped virus with a positive-sense single-stranded RNA as a typical characteristic [116]. This contagious virus utilizes a spike (S) protein to bind to the angiotensin-converting enzyme 2 (ACE 2) receptor on the cellular membrane and the Transmembrane Serine Protease 2 molecule, and then cleaves the S protein to allow the viral membrane to fuse with the cellular membrane where viral RNA enters the cell [117]. Thus, anywhere that ACE 2 receptors are expressed, such as the endothelial cells or smooth muscle cells, in several organs, and in the nasal epithelium, is vulnerable to SARS-CoV-2, which can lead to multiorgan failure [21]. However, once inside the cell, the uncoated RNA translates polyproteins (pp1a and pp1ab), which replicate and transcribe more viral RNA complexes into virus-induced double-membrane vesicles [116]. SARS-CoV-2 is a virus with devastating consequences compared to other human coronaviruses, and its effects on certain vulnerable populations, such as immunocompromised people or pregnant women, have been significant during the ongoing pandemic and will be after our transition to endemic status [21].

### 3.1. SARS-CoV-2 and Pregnancy

SARS-CoV-2 has been indiscriminate in its ability to infect the world population; however, those that are more vulnerable, such as pregnant women, have found the emergence of this new virus particularly stressful and impactful. These new mothers have experienced an increase in negative birth outcomes, as 23.6% of neonates were born preterm, and 86% of these births were delivered by cesarean sections, where the premature rupture of membranes (PROM) and fetal distress were among the indicated causes [118]. Studies have shown that pregnant individuals who are positive for SARS-CoV-2 can transfer the infection to multiple sites that are essential for a healthy pregnancy; these include the placenta, amniotic fluid (AF), and breastmilk (Figure 2). Unfortunately, there are only a few studies that aim to understand the transmission of SARS-CoV-2 virus through the tissues of pregnancy to the fetus. However, it has been shown that the placental villi can become infected with SARS-CoV-2, with or without the transmission of infection to the fetus [119]. Specifically, it was seen that the expression of SARS-CoV-2 was found consistently on the syncytiotrophoblast [119]. Others who have performed systematic reviews have focused on understanding the transmission rate to the breast milk, AF, cord blood, and placenta, although these were found at low rates in the majority of the studies (Figure 2) [120]. This indicates that we have much to understand as to why, in some pregnancies, the ubiquitous transmission of the virus occurs, whereas, in others, the tissues of pregnancy and the fetus are spared.

Given the short duration of this pandemic (2 years), the limited amount of data available is not surprising. Thus, further studies that focus on improving our understanding of SARS-CoV-2 in pregnancy are critical to improve the increased negative outcomes of PTB and stillbirth when the mother is infected. In addition to the delivery risks associated with this virus, one of the most significant conditions that mothers are at an elevated risk for is preeclampsia. This is strongly associated with SARS-CoV-2, and this is independent of any risk factors or preexisting conditions. Indeed, it has been demonstrated that nulliparous women are particularly at risk [121]. This risk is notable in that it has also been shown that non-symptomatic women with a SARS-CoV-2 diagnosis are also at an increased risk for maternal morbidity and preeclampsia [122]. Therefore, pregnant mothers should be encouraged to follow SARS-CoV-2 preventative measures.

A comprehensive table of publications that illustrate what is understood about RAGE/AGE in pregnancy is provided in Table 1; however, it is notable that the specific role of RAGE in pregnancy with SARS-CoV-2 has yet to be studied. It is also important to understand how socioeconomic status, geographic location, access to nutrition and healthcare, and the key comorbidities influence the health of the mother and fetus in the climate of the pandemic (Figure 1).

### 3.2. Pregnancy Is an Additional Physiological Challenge That Can Exacerbate the Severity of SARS-CoV-2

While pregnancy is not a comorbidity, the increased risk of severe SARS-CoV-2 disease, and the increase in both obstetric complications and negative fetal health outcomes in pregnant women, place it within the category of underlying conditions that exacerbate this virus. Early in the pandemic, it was seen that the disease course and outcomes were similar in pregnant and nonpregnant women; this was when the disease studied was mild. However, subsequently, an increase in hospitalizations and intensive care admissions has been seen for pregnant patients [123]. When compared to individuals without SARS-CoV-2 infection, infected pregnant individuals have significantly increased risks of preterm delivery, negative neonatal outcomes, and maternal death (approximately 17% higher, due to pregnancy-related hypertension and postpartum hemorrhage [124,125,126]). Additionally, data from a study performed in Scotland, the United Kingdom, show that 100% of all fetal deaths recorded, and 90% of intensive-care-unit admissions, came from individuals that were not vaccinated for SARS-CoV-2 [126]. Thus, while it cannot definitively be concluded that SARS-CoV-2 is always more severe in pregnancy, it has been established that there are legitimate health concerns in instances where the mother has one (or more) of the comorbidities that are known to worsen SARS-CoV-2 symptoms, as they have higher risks of severe disease progression and negative neonatal outcomes (Figure 1) associated with excessive RAGE-mediated inflammation [22].

### 3.3. SARS-CoV-2 Vaccination

At present, the vaccination rate (for those with two vaccinations) in the United States is 66.1%, with the most commonly utilized vaccines being Pfizer, Moderna, and Johnson&Johnson [127]. However, the rapid implementation of these vaccines seems to be contributing to the hesitancy of new mothers to get vaccinated. Other contributing factors seem to be driven by misinformation, and the perceived lack of data on how vaccines affect the fetus and the mother during gestation [128,129]. Despite this, pregnant women are encouraged to take advantage of the available vaccines because of their increased risks of severe disease and the increase in negative outcomes that have been documented since the pandemic began [130]. In further support of vaccination, pregnant women are able to pass antibodies to the fetus through the placenta and breastmilk [131].

## 4. The Interaction of RAGE and SARS-CoV-2

Although SARS-CoV-2 was only discovered at the end of 2019, its connection with RAGE has already been established [8,17,22]. However, due to its most dramatic effects in patients with already established comorbidities, this is where we currently have the greatest level of understanding of the role of this receptor [8,17,22]. Indeed, much less is understood about the role of RAGE in SARS-CoV-2 cases that are not complicated with comorbidities. Thus, what is currently understood is that the expression of RAGE in adult lungs implicates it in the disease inflammatory pathogenesis [22], where its hyperactivation has also been linked with other organ diseases, including inflammatory bowel disease [33], neurological inflammation [132], and worse outcomes in the elderly [8].

As this virus replicates itself and infects cells throughout the body, the symptoms can range from person to person, depending on their specific personal vulnerabilities, taking the form of mild to severe disease. For those with severe SARS-CoV-2, the symptoms of ARDS or pneumonia are common, leading to a high risk of reduced lung capacity and hospitalization, where negative outcomes (i.e., septic shock and multiorgan failure) and invasive treatments (i.e., oxygen therapy, mechanical ventilation, and dialysis) are more likely to be required [3,133]. Infected individuals with severe symptoms have been shown to have an overexpression of RAGE in both inflammatory and endothelial cells, as compared to those with mild symptoms of SARS-CoV-2 [22]. Patients with these severe symptoms are also more likely to have a range of comorbidities, which have been shown to already to have an increased level of RAGE expressed, and, thus, with SARS-CoV-2 infections compounding their preexisting issues, may, as a result, hyper-stimulate RAGE [22,133].

### 4.1. RAGE and Its Role in Comorbidities Associated with SARS-CoV-2

Due to the current pandemic of SARS-CoV-2, and the inevitable transition to endemic status, this virus and its associated features have been intensely studied. These data have illustrated that, of the >3 million deaths attributed to the virus, 98% are from individuals that have one or more comorbidities, such as DM, obesity, hypertension, chronic lung disease, and CVD [134]. Furthermore, it has been shown that these comorbidities directly correlate with, and dictate, the severity of the illness [134]. Whilst not a comorbidity itself, pregnancy does seem to mimic this phenomenon and, in many cases, increases the risk of severe disease and therefore the SARS-CoV-2 intensity. Inflammatory cytokine production from the comorbidities or pregnancy establishes a higher baseline of inflammation that involves RAGE mechanisms, which will increase further if the patient has to fight additional inflammation resulting from SARS-CoV-2 infection. Although this needs further study, increasing our understanding of how all these factors combine, or interact, could enable the use of the sRAGE levels as a predictive measure for the severity and potential lung damage caused by SARS-CoV-2 infection.

#### 4.1.1. Diabetes and Obesity

The imbalance in the regulation of glucose in the body in types 1 and 2 DM can lead to potentially life-threatening conditions, such as ketoacidosis, CVD, and nephropathy [17,22]. A common symptom, hyperglycemia, also contributes to the formation of AGEs, which are produced from glucose, as well as other intermediates of glycolysis, such as methylglyoxal, which is a precursor to several AGEs [8]. As AGE accumulation increases, so do their interactions with the principal receptor RAGE. This activates the cascade of proinflammatory cytokine production, which thus initiates an innate immune response.

Obesity follows a similar pattern of immunocompromisation in the form of endothelial dysfunction, hypercoagulability, and thrombosis. However, the AGE accumulation in obesity can be attributed to the interaction of adipose tissue with RAGE, leading to adipokine secretion. RAGE also increases the activation of adipose tissue macrophages [135]. The secretion of the DAMP HMGB1 by adipocytes fosters a cycle of immune cell stimulation and adipocyte necrosis, which leads to chronic inflammation and reactive-oxygen-species (ROS) production. These conditions not only build an environment that is beneficial for the development of DM and CVD, but they also benefit SARS-CoV-2 infection through the high ACE2 expression found in adipose tissues. Although not fully understood, a link between the adipocyte expression of ACE2 and SARS-CoV-2 infection could explain, in part, the multiorgan damage that is seen in more severe cases. It is also thought that the accumulation of DAMPs in obesity and diabetes would, in part, through RAGE, lead to the increased risk of severe SARS-CoV-2 [136]. Thus, these individuals are subject to what has been described as ‘inflammatory priming’ (Figure 1) [137]. In addition, pregnancy appears to amplify this, as those who were obese before becoming pregnant had more severe SARS-CoV-2 disease [138].

#### 4.1.2. Hypertension and Pulmonary Disease

Consistent with the previous comorbidities, hypertension is also associated with severe SARS-CoV-2 and higher mortality rates. Additionally, its formation can be linked to the interaction of RAGE and AGEs, which stimulates ROS and proinflammatory cytokine production while subsequently exerting negative effects on the bioavailability and activity of nitric oxide [22,139]. In this manner, arterial stiffness and vascular resistance are modulated by the accumulation of AGEs and their receptor. The relationship between these molecules and hypertension can be further described by a direct correlation with plasma AGE, as well as by an indirect correlation with sRAGE [22].

RAGE’s involvement in pulmonary disease is established in both noninfectious and infectious etiologies. It can be characterized by an increase in RAGE and its ligands in response to lung epithelial damage, which forms from conditions such as ARDS, pneumonia, and fibrosis. The dominant and widely understood outcome of SARS-CoV-2 infections is the wide range of associated lung damage that occurs. This is attributed to its infectious pathway, as the SARS-CoV-2 particles bind by their surface spike protein to ACE2 on cellular surfaces. Along with its expression in adipocytes, ACE2 is commonly expressed in types I and II alveolar epithelial cells, kidney tubules, smooth muscle, and endothelial cells [34,136]. Alveolar epithelial cells, which also exhibit elevated RAGE expression, therefore play an important role in the onset of ARDS, which is also critically associated with SARS-CoV-2 death [8].

## 5. SARS-CoV-2 and Its Impact on Hawai’i and Its Vulnerable Populations

Although SARS-CoV-2 has affected everyone, due to the propensity of the comorbidities that influence infection outcomes, this virus has had its largest impact on specific populations. This has drawn our attention to the inequity in the access to healthcare and the ability of minority communities to recover from infection. One such example of this has been seen in Hawai’i. This is because the comorbidities that influence both pregnancy outcomes and SARS-CoV-2 severity are prevalent in Hawai’i. Heart disease is the leading cause of death, diabetes is the fifth leading cause of death, and Native Hawaiians and Pacific Islanders are 80% more likely to be obese compared to non-Hispanic Caucasian people [140,141,142]. Therefore, if any woman had any of these comorbidities and became pregnant, she would be at a high risk for severe SARS-CoV-2. In Hawai’i, the rates of SARS-CoV-2 infection and mortality are lower when compared to the United States. However, the socioeconomic challenges and the states’ authoritative response to the virus have also contributed to the health outcomes of the islands. As of April 2022, in the United States, 80.53 million people had been infected, with 988,545 deaths. However, in Hawai’i, there have been only 240,000 cases with 1406 total deaths [143]. As a small chain of islands in the Pacific that is the most geographically isolated place in the world, the multiethnic and large cultural diversity has played a huge role in the infection rates in the different communities. Universally, it is very clear that those of lower socioeconomic status are more susceptible to SARS-CoV-2. However, in Hawai’i, this is tragically often Native Hawaiians and Pacific Islanders.

The picture for the Pacific Islander (i.e., Samoa and Micronesia) communities is even more challenging. Although this community only comprise about 4% of the total population, they have accounted for 31% of the total reported SARS-CoV-2 cases in Hawai’i [144]. Thus, together, Native Hawaiians and Pacific Islanders have the highest SARS-CoV-2 death rates compared to any other ethnic group [145]. It is thought that this may be, in part, due to the numerous socioeconomic barriers that they face, including living in multigenerational housing, with a big emphasis on family as central to their cultural values, which has made it more difficult to isolate and socially distance during this pandemic [144]. Other barriers, such as a lack of access to education, nutrition, and healthcare, language barriers, and being underinsured, are some of the issues that this group has consistently faced, both before and during the SARS-CoV-2 pandemic [146,147,148,149].

### Hawai’i and Pregnancy

Many of the native peoples of Hawai’i and the Pacific have suffered significantly during the pandemic, and, therefore, this pandemic has been even more onerous for those who have been pregnant. The current dataset on the SARS-CoV-2 infection and vaccination rates in Hawai’i is severely lacking, as there is no breakdown of the data in terms of pregnancy. This paucity of information availability only fuels vaccine hesitancy in pregnant women, as it makes it difficult for lay individuals to determine the best choice. However, on the basis of the understanding that pregnant women with comorbidities are at a higher risk for negative pregnancy outcomes, and that, if they contract SARS-CoV-2, this is further increased, this is a reality that is particularly concerning for Hawai’i. Here, 1 in 10 babies are born preterm, and this is influenced by risk factors, such as the fact that 23.2% of the child-bearing-age population is obese, and Native Hawaiian and Pacific Islander mothers are at 39% higher risk for PTB compared to Caucasian mothers [150]. However, this is obviously compounded in this population if they have any of the comorbidities that increase the risk for severe SARS-CoV-2 [150,151]. Furthermore, social inequities, such as access to healthcare, are an issue for many pregnant women in Hawai’i, as 1 in 17 child-bearing-aged women are uninsured [15,42]. Thus, there is a need to collect more comprehensive data to determine the key factors that influence the health of the pregnant population in Hawai’i, and to therefore improve the severe disparities that are faced by our Indigenous communities, and especially while we are battling the ongoing pandemic. This is an exemplar community at risk, and there are many others who are suffering similarly within their own geographical or national contexts. Thus, there is a great need for scientific attention to understand what is happening to those most at risk, and for the development of polices that address health inequities. 

## 6. RAGE as a Biomarker or Putative Therapeutic Target

Given the current gaps in our knowledge, for those with severe SARS-CoV-2 with dangerous comorbidities who may also be pregnant, as well as vulnerable minorities, there is a legitimate need for further research into the unifying molecule RAGE. This may help us to understand its potential as a biomarker and for therapeutic strategies, including those for use in pregnancy. Although the pathways that regulate sRAGE and its associated roles are not completely understood, the measurement of the relative levels of different species of RAGE expression have proven useful as a biomarker in other conditions. Multiple polymorphisms for the receptor have also been identified; however, the most studied, Gly82Ser, is a single-nucleotide polymorphism within the RAGE gene that modifies the ligand-binding structure by increasing its binding affinity for AGEs. This genotype is characterized by lower sRAGE levels and higher risk factors for CVD, which has implicated it in diabetes and autoimmune diseases, such as Kawasaki disease [25]. Thus, abnormal upregulation and sRAGE activation are indicative of disease development and inflammatory and immune responses in the body [28,103,152,153]. Generally, increased levels of membrane-bound RAGE are associated with disease pathogenesis/injury because the ligands bound to fl-RAGE are able to transduce that signal intracellularly and stimulate the nuclear translocation of NF-κB, thus producing inflammation. In contrast, increased levels of sRAGE indicate a higher instance of blocked ligand binding, which therefore prevents the activity of membrane-bound RAGE and its downstream signaling. However, this is not always the case, as different levels of disease progression correlate with different changes in sRAGE. For example, coronary artery disease and diabetes complications are associated with decreases in sRAGE, whereas an increase in sRAGE is seen in instances of impaired renal function and with end-stage disease [29]. In addition to the many studies implicating RAGE activation and/or the sRAGE circulating levels in disease progression, recent studies have shown that RAGE signaling is also activated in instances of SARS-CoV-2 infection, and particularly when the patients have comorbidities. Although not fully understood, this adds significant value to this receptor as a diagnostic tool that could direct treatment strategies and help to predict the inflammatory disease severity [30,154,155]. Current therapeutic strategies aim to disrupt RAGE signaling through the direct inhibition of RAGE and/or its ability to interact with its ligands via medicines such as chloroquine, thrombomodulin, or even soluble ACE2 [17]. HMGB1 is the RAGE ligand that is most studied in pregnancy that could be used to lower SARS-CoV-2 through the anti-agents heparin, resveratrol, or metformin [156]. The goal of these therapies is to ultimately disrupt the inflammatory pathways that correlate with disease progression. Because of the complexity and critical association with multiple signaling pathways, the need for further study and an understanding of the RAGE system should be a priority.

## 7. Concluding Remarks

The complex nature of the interaction between disease states, risk factors, and ethnicity are difficult to separate to understand their relative contributions to health. However, when they all coalesce on a specific pathway(s), it is tempting to postulate that this could be leveraged into useful tools to improve health for vulnerable populations, such as those in Hawai’i. The seemingly additive nature of the health challenges of hypertension/CVD, DM, and obesity, with either a pregnant state and/or SARS-Co-V2, leads to increasingly bad health outcomes. RAGE has been implicated in all of these; thus, it could be a unifying receptor and inflammatory driver. However, this has not been directly studied, and so, although inflammation generally seems to be additive, leading to phenomena such as cytokine storms, it is not known if the RAGE system reaches a maximum, or if it continues to work in a feedforward manner as you continually activate the proinflammatory systems. Thus, as RAGE has been measured as a biomarker in other diseases [17,30,139,154,155], it may be useful to predict how severe the SARS-CoV-2 infection will be, with and without pregnancy. Thus, it has the potential to be useful as a prognosis driver, as the higher its activation, the worse the outcome, and the more aggressively conditions might be treated.

As RAGE has an intrinsic sRAGE inhibitor mechanism, this could be leveraged to increase the sequestering of RAGE ligands, either through the synthesis of artificial mimetics or the augmentation of the endogenous system. Indeed, it is not clear whether sRAGE crosses into the fetus in normal pregnancy. Another avenue to choreograph RAGE-induced inflammation could be to manipulate the availability of the γ-secretase system that cleaves fl-RAGE to sRAGE. However, legitimate concerns to this approach in pregnancy might be centered around the possibility that these ligands could cross the placenta and affect fetal development, and especially given that RAGE is fairly ubiquitously expressed across tissues during development.

The potential of RAGE as both a future biomarker and putative therapeutic target invites optimism for those most at risk from pregnancy complications, which now include SARS-CoV-2. The Hawai’i-specific context and its vulnerable populations have directed our thinking about the interaction and complexity of this contemporary burden, as we learn to live with SARS-CoV-2 and the transition to endemic status. However, this is also relevant for other vulnerable populations in other locales, as the disproportionate rates of these health disparities for the Indigenous populations in Hawai’i are paralleled by underrepresented and underserved communities globally.

## Figures and Tables

**Figure 1 ijms-23-06359-f001:**
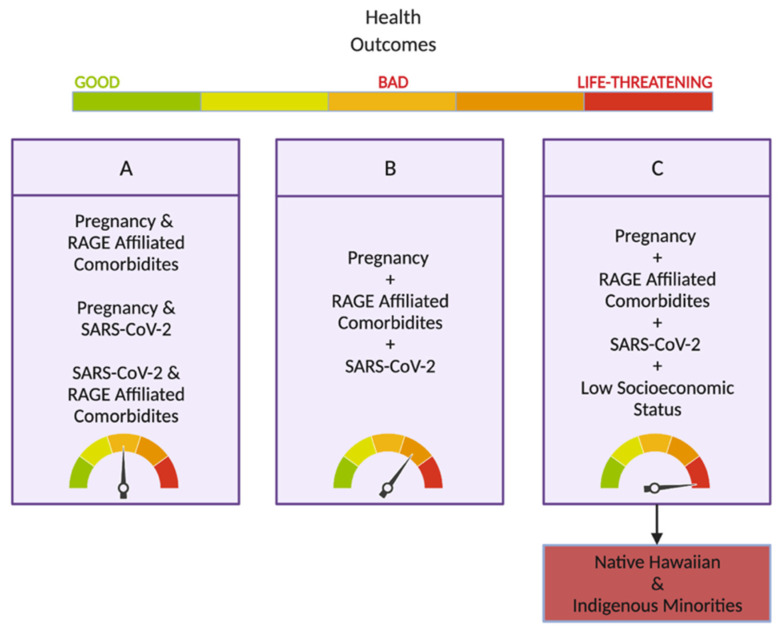
The Synergistic Influence of RAGE-Affiliated Afflictions on Negative Health Outcomes. Three different scenarios contributing to the occurrence of negative health outcomes. Scenario A depicts the coexistence of two conditions in an individual leading to higher risk of poor/bad health outcomes. Scenario B describes an individual with three concurrent conditions and the further increased rate of poor health outcomes due to the accumulation of risk factors. Scenario C builds on the conditions established in Scenario B by adding the context of a low socioeconomic status. This further aggravates the negative synergistic effects and creates an environment conducive to severe health outcomes, which is unfortunately representative of many Native Hawaiian and Indigenous minority populations in the Pacific. The figure was created with Biorender.

**Figure 2 ijms-23-06359-f002:**
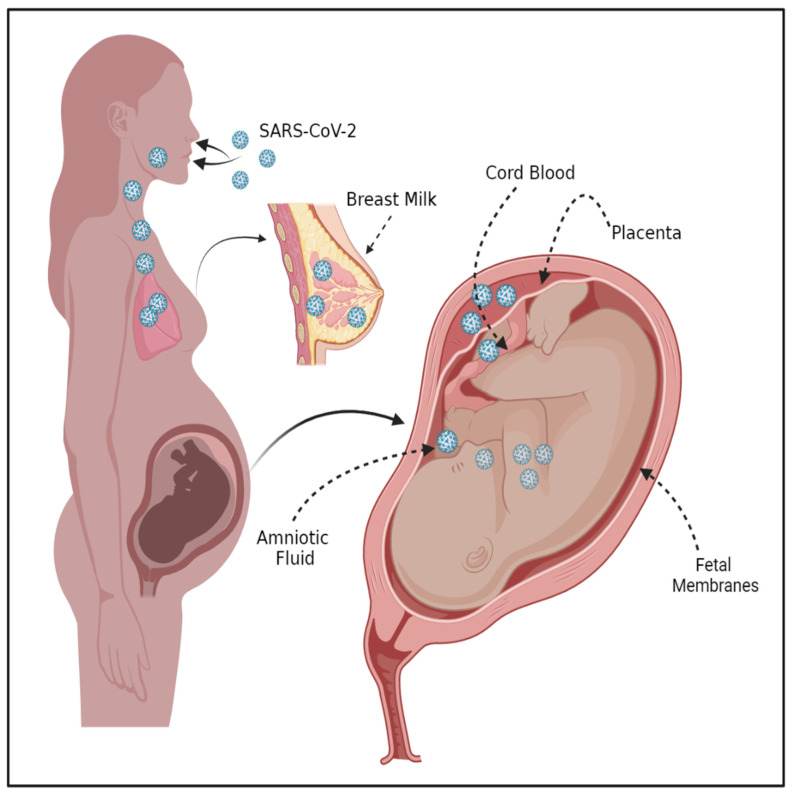
The Vertical Transmission of SARS-CoV-2 from the Maternal to Fetal Compartment. SARS-CoV-2 virus particles enter via the mouth and nose and travel both throughout the vascular system and to the uterus. Positive expression of SARS-CoV-2 has been documented in the placenta, cord blood, and amniotic fluid. The fetus can also then be colonized by SARS-CoV-2. The figure was created with Biorender.

## Data Availability

All of the data used in this literature review is publically available.

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
