# Peer review of "RAGE against the Machine: Can Increasing Our Understanding of RAGE Help Us to Battle SARS-CoV-2 Infection in Pregnancy?"

_ijms, 2022, doi:10.3390/ijms23126359_

Round 1
Reviewer 1 Report
The authors reviewed the literature about the potential role of RAGE activation / signaling in SARS-Cov-2 infection in pregnancy, with a focus on the local situation (Hawai'i). The manuscript presents no major flaws. However, a few points should be addressed.
1. Fig. 1 was positionated after Fig. 2 in the text. Why?
2. Lines 82-83. The words "S100B proteins, EN-RAGE (S100A12)" should be replaced by the words "certain S100 proteins" as S100B and S100A12 are not the sole S100 proteins able to activate RAGE (see PMID: 18331229).
3. The concept expressed in lines 261-272 should be expaned and supported by appropriate Refs. (for example see PMID: 33327744; PMID: 32892690; PMID: 34392108).
Author Response
The authors reviewed the literature about the potential role of RAGE activation / signaling in SARS-Cov-2 infection in pregnancy, with a focus on the local situation (Hawai'i). The manuscript presents no major flaws. However, a few points should be addressed.
- Fig. 1 was positioned after Fig. 2 in the text. Why?
We apologize for this error. The journal editors entered the figures into the text template and we are not sure why they did this. We are sorry for any difficulty or confusion this caused during your review of our article.
- Lines 82-83. The words "S100B proteins, EN-RAGE (S100A12)" should be replaced by the words "certain S100 proteins" as S100B and S100A12 are not the sole S100 proteins able to activate RAGE (see PMID: 18331229).
This has been changed in the text. Thank you for the suggestion and reference.
- The concept expressed in lines 261-272 should be expanded and supported by appropriate Refs. (for example see PMID: 33327744; PMID: 32892690; PMID: 34392108).
Thank you for this suggestion, we reviewed the references you provided and expanded the paragraph and feel it is greatly improved. We hope you agree this is sufficient.
Reviewer 2 Report
In the review entitled "RAGE against the machine: Can increasing our understanding of RAGE help us to battle SARS-CoV-2 infection in pregnancy?" the authors suggest a possible link between the receptor RAGE and Sars-CoV-2 infection, in pregnancy. RAGE has been linked in different reviews to SARS-CoV-2 infection, in particular, high levels of RAGE and its ligands, which characterized several SARS-CoV-2 comorbidities (such as diabetes, hypertension, etc), might aggravate SARS-CoV-2 infection. To my knowledge, this is the first review in which the receptor RAGE has been associated with SARS-CoV-2 infection in pregnancy, and it could be very interesting. Nevertheless, is necessary to deeply investigate some aspects and clarify this possible link.
- The authors should add a paragraph about pregnancy and susceptibility to infection
- The authors should deeply investigate the role of RAGE and its ligands in pregnancy, and in infection during pregnancy. Moreover, could be interesting a paragraph about the role of RAGE, and also of SARS-CoV-2, in preeclampsia.
- In the paragraph "SARS-CoV-2 in pregnancy" the first part (from line 156 to line 181) should be inserted into a separate paragraph concerning SARS-CoV-2 in general. Instead, in this paragraph, the role of SARS-CoV-2 in pregnancy should be deepened.
- Many reviews have already investigated the possible link between RAGE and SARS-CoV-2, especially in comorbidities, please summarise paragraph 3.1.
- In my opinion, paragraphs 4.1 and 4.2 should be eliminated, and some of this informations should be inserted in the conclusions.
- Paragraph 5 is not clear, where is the connection between RAGE and SARS-CoV-2 in pregnancy?
Author Response
In the review entitled "RAGE against the machine: Can increasing our understanding of RAGE help us to battle SARS-CoV-2 infection in pregnancy?" the authors suggest a possible link between the receptor RAGE and Sars-CoV-2 infection, in pregnancy. RAGE has been linked in different reviews to SARS-CoV-2 infection, in particular, high levels of RAGE and its ligands, which characterized several SARS-CoV-2 comorbidities (such as diabetes, hypertension, etc), might aggravate SARS-CoV-2 infection. To my knowledge, this is the first review in which the receptor RAGE has been associated with SARS-CoV-2 infection in pregnancy, and it could be very interesting. Nevertheless, is necessary to deeply investigate some aspects and clarify this possible link.
- The authors should add a paragraph about pregnancy and susceptibility to infection
Thank you for this suggestion, we think this addition really improves the paper. In response we have added more in the narrative and it is also highlighted in Table 1.
- The authors should deeply investigate the role of RAGE and its ligands in pregnancy, and in infection during pregnancy.
We have added a comprehensive table (Table 1) that organizes the published articles on RAGE/AGE in pregnancy. We hope you agree this is sufficient to address your concerns.
Moreover, could be interesting a paragraph about the role of RAGE, and also of SARS-CoV-2, in preeclampsia.
An increased narrative has been included regarding the risks of PE. Thank you for this suggestion we agree this really adds to the article.
- In the paragraph "SARS-CoV-2 in pregnancy" the first part (from line 156 to line 181) should be inserted into a separate paragraph concerning SARS-CoV-2 in general. Instead, in this paragraph, the role of SARS-CoV-2 in pregnancy should be deepened.
We have separated out these topics and expanded the pregnancy paragraph to help improve and deepen this section. We hope you agree with us that this improves the article.
- Many reviews have already investigated the possible link between RAGE and SARS-CoV-2, especially in comorbidities, please summarize paragraph 3.1.
Many of these are littered throughout the article, as there are not many (17 in total in pubmed), we decided to add a few more and hope you agree that their use in other sections strengthens the substantiation of those paragraphs and that you agree we have been thorough as we have cited nearly all of them.
- In my opinion, paragraphs 4.1 and 4.2 should be eliminated, and some of this information should be inserted in the conclusions.
We have kept these sections, as we believe this is an important commentary to provide our home context to these issues. We believe our population vulnerabilities will provide and exemplar population that others will read about and use as a prompt to review the vulnerabilities in their own specific populations more carefully. We hope we have made our intention in the manuscript clearer now, and that you agree this is an important opportunity to draw attention to the inequities that have become even more apparent due to the pandemic.
- Paragraph 5 is not clear, where is the connection between RAGE and SARS-CoV-2 in pregnancy?
Thank you for allowing us to improve this paragraph and we now hope you agree this connection is clearer.
Reviewer 3 Report
In the review paper “RAGE against the machine: Can increasing our understanding of RAGE help us to battle SARS-CoV-2 infection in pregnancy?” the authors summarized the role of RAGE receptor as a key driver of inflammation in pregnancy and in comorbidities that are known to aggravate COVID-19 especially if combined with pregnancy.
Comments
- English should be improved; e.g., in the abstract “RAGE is known to bind a number of ligands produced by tissue damage and cellular stress, thus its activation results in the activation of the pro-inflammatory transcription factor Nuclear Factor Kappa B (NF-κB) and the subsequent generation of key pro-inflammatory cytokines” (lines 9 – 11) repeating same word (activation) is not necessary.
It would sound better saying: thus its activation triggers pro-inflammatory transcription factor Nuclear Factor Kappa B (NF-κB) and……..
Or: “The comorbidities of hypertension, cardiovascular disease, diabetes and obesity are known to lead to poor pregnancy outcomes and have also been shown to have high RAGE activation when these individuals are infected with SARS-Co-V-2 (lines 13-14)”.
Instead, I propose: ….. have also been linked to RAGE activation when these individuals are infected with SARS-Co-V-2; etc…
Or: “A healthy term pregnancy is dependent on the interplay of an incredible number of factors, ….” (line 22). I suggest changing word incredible to numerous….
And it’s going on……….
- It is strange that Figure 2 is coming before Figure 1. In Figure 1, there is no explicit abbreviation for SES, which I assume is socioeconomic status; it should be stated in the Legend.
- I do not think that pregnancy should be called comorbidity in any context, and especially not in subtitle: 1.3. Pregnancy as a Comorbidity of SARS-CoV-2 (line 293)
- There is nothing in the title or abstract indicating that short analysis of situation of COVID-19 combined or not with pregnancy in Hawaiʻi is done. This analysis is important but it’s mentioned in the review out of the blue.
- Table reviewing articles about the connection of RAGE and pregnancy with or without COVID-19 should be included.
- Figure 1 is titled “The Synergistic Influence of RAGE Affiliated Afflictions on Negative Health Outcomes”. Yet no RAGE is included anywhere in the Figure.
- Figure proposing the role of RAGE in SARS-CoV-2 infection in pregnancy is necessary.
Author Response
Thank you so much for taking the time to provide these suggestions, we appreciate the effort this took. We have made these alterations to the text and many others and hope you agree that it is now improved.
- It is strange that Figure 2 is coming before Figure 1. In Figure 1, there is no explicit abbreviation for SES, which I assume is socioeconomic status; it should be stated in the Legend.
- Figure 1 was placed in the document by the journal editors not the authors and so we apologize for any confusion this caused during review.
SES has been spelled out in full in the figure.
- I do not think that pregnancy should be called comorbidity in any context, and especially not in subtitle: 1.3. Pregnancy as a Comorbidity of SARS-CoV-2 (line 293)
- Thank you for this comment we agree and struggled with how to describe this physiological challenge. We hope you agree that we have more suitably described the additional physiological stress this causes, including when the mother has SARS-CoV-2.
- There is nothing in the title or abstract indicating that short analysis of situation of COVID-19 combined or not with pregnancy in Hawaiʻi is done. This analysis is important but it’s mentioned in the review out of the blue.
We agree that this analysis is important and illustrates our specific context for these issues, and provide an exemplar at risk population. We have mentioned that this population is at risk in the abstract to indicate that they will be discussed in the review.
- Table reviewing articles about the connection of RAGE and pregnancy with or without COVID-19 should be included.
- This table has been added and we thank you for this suggestion, as we agree this greatly improves the article. It also provides the opportunity for us to point out that RAGE, pregnancy and COVID-19 have not been studied in concert.
- Figure 1 is titled “The Synergistic Influence of RAGE Affiliated Afflictions on Negative Health Outcomes”. Yet no RAGE is included anywhere in the Figure.
The Figure has been amended to make this clearer.
- Figure proposing the role of RAGE in SARS-CoV-2 infection in pregnancy is necessary.
We have included a comprehensive table and a longer narrative regarding this, and therefore have not also included an additional figure. We hope you agree this is sufficient.
Round 2
Reviewer 1 Report
The authors have satisfactorily addressed my concerns.
Author Response
Thank you so much for the time and effort you took to review our manuscript. We are thankful for your suggestions and agree they made it a stronger paper.
Reviewer 2 Report
The authors performed several improvements in the revised version of the paper, but, in my opinion, some changes are still needed.
The authors should insert a separate paragraph about pregnancy and susceptibility to infection, and the role of RAGE, and its ligands, in infection during pregnancy, deepening the data reporting in table 1.
Tab 1 is very confused. The authors should: i) use the full name of the pathologies in the first column; ii) divide human, animal, and in vitro data; iii) eliminate the lines referred to as "review"; iv) eliminate lines of "Non-Pathogenic Pregnancy", "general", and "General Review"; v) replace "infection" with "Infection during pregnancy"; vi) eliminate all the "No association" data. vii) replace "Genomics" and "gene" with a specific tissue in "Tissue Targets Studied" column, and change the title of this column with "Tissue or cell Targets".
Author Response
Thank you so much for taking the time to so thoroughly review the table, we agree with all of your suggestions and hope that you agree that this makes the table much less confusing. It really adds to the value of the manuscript.
Reviewer 3 Report
The manuscript has been considerably improved making this review paper more appealing to the audience and more relevant to the field. Few minor comments:
- Letters in Table 1 should be bigger, would be easier to read.
- Legend for the Table is usually above the Table (in contrast with the Figures).
- English should be improved.
Author Response
We have increased the font size to hopefully strike a balance between the table taking up a lot of space in the manuscript but also ensuring that it is easy to read.
We have moved the table legend to before the table.
We have reedited the document for English language and grammar and hope you agree that it is now reached a publishable standard. Thank you for your time and effort in reviewing our manuscript, all of the suggestions you mad were most useful and we believe have helped us improve our review.